# Structural Cohesion, Role Equivalence, or Homophily: Which Process Best Explains Social Homogeneity?

**DOI:** 10.3390/ijerph192114471

**Published:** 2022-11-04

**Authors:** Ignacio Ramos-Vidal

**Affiliations:** 1Department of Social Psychology, University of Seville, 41018 Sevilla, Spain; iramos5@us.es; Tel.: +34-955554343; 2Instituto Universitario de Estudios sobre América Latina, University of Seville, 41004 Sevilla, Spain

**Keywords:** heterophily, homophily, psychosocial intervention, role equivalence, sense of community, social network analysis, social homogeneity, structural cohesion

## Abstract

Social homogeneity, understood as the similarity of perceptions and attitudes that individuals display toward the environment around them, is explained by the relational context in which they are immersed. However, there is no consensus about which relational mechanism best explains social homogeneity. The purpose of this research is to find out which of the three classical relational processes most studied in network analysis (structural cohesion, role equivalence, or homophily) is more determinant in explaining social homogeneity. To achieve the research objective, 110 professionals (psychologists, social workers, and community facilitators) implementing a psychosocial care program in three regions of Northwest Colombia were interviewed. Different types of relationships among professionals were analyzed using network analysis techniques. To examine the structural cohesion hypothesis, interveners were categorized according to the level of structural cohesion by performing core-periphery analysis in the networks evaluated; to test the role equivalence hypothesis, participants were categorized according to their level of degree centrality in the networks examined; to test the homophily hypothesis, participants were grouped according to the level of homophily in terms of professional profile. The non-parametric tests showed that role equivalence was the most powerful mechanism for explaining social homogeneity in the sample of psychosocial interveners evaluated.

## 1. Introduction

Understanding the processes which condition the structure and dynamics of groups is a core question for social scientists. Empirical evidence highlights that many psychosocial phenomena that determine individual behavior are, at least partially, explained by the relational context in which individuals are immersed. One of the main objectives of the study in social and behavioral sciences is to understand the factors that determine the development of attitudes, which, in turn, are essential for understanding human behavior [1,2]. An attitude is a precise or abstract evaluation of a positive or negative nature, which may refer to individuals, objects, events, activities, or ideas [3]. Some authors suggest that attitudes are psychological tendencies that individuals express by evaluating a particular entity with a certain degree of positivity or negativity [4]. Social perception refers to the process of identifying and using social cues that enable individuals to make judgments about roles, norms, relationships, and interaction contexts [5]. The study of the factors that contribute to individuals developing similar attitudes and perceiving their social context in a similar way is central since both processes determine behavior [6,7].

Understanding the social forces that cause individuals to develop similar attitudes and perceive the social environment is one of the topics of greatest interest to social scientists [8]. According to Gehrlein [9], social homogeneity describes the extent to which individual preferences tend to be similar, or, in other words, that they develop a similar view of the world around them. Since the mid-1950s, there have been numerous proposals aimed at understanding the psychosocial processes that cause similarity when evaluating the same social context [10]. Some of these initiatives suggest that individual attributes are the main triggers for the concordance of attitudes, perceptions, and behavior by members of the same social system. Therefore, this approach emphasizes that the similarity of actors’ characteristics is the factor that best explains why individuals have the same attitudes and evaluate the social context in which they are immersed in the same way. From this logic, different studies suggest that individual characteristics predict attitudes toward paid work [11], volunteering [12], immigration [13], political parties [14], or homosexuality [15], among many other phenomena. Other approaches postulate that social homogeneity does not depend exclusively on individual variables but on the social forces operating and interacting in each social environment. The classic study on electoral behavior by Lazarsfeld, Berelson, and Gaudet [16] showed that the decision regarding which political party to vote for depends to some extent on the social contacts of individuals. Another seminal study supporting this reasoning was Coleman, Katz, and Menzel’s [17] research on the adoption of a new drug (tetracycline) by doctors. This research highlighted how doctors who interacted, both professionally and informally, with other doctors who had administered the new drug to their patients were more likely to adopt the medical innovation. Although some studies suggest that aggressive marketing strategies are the main predictor of tetracycline adoption [18], the impact of social contagion on the decision to adopt the innovation seems difficult to refute. From a similar perspective, research applying social network analysis (SNA) shows that the mechanisms of social influence that come into play through interpersonal relationships determine both the formation of attitudes and perceptions and the behavior that individuals perform [19].

The methodological progress made by SNA has contributed to understanding the multiple effects that social structures have on behavior. Erickson [20] proposes that the configuration of attitudes and perceptions, as well as their resulting impact on behavior, have a solid relational basis. The author points out that the natural unit of analysis to study this phenomenon is social networks rather than isolated individuals and that the main phenomenon that explains the formation of attitudes is the degree of consensus shown by a group of individuals occupying the same position within a social structure. Thus, interpersonal relationships and the social structure underlying them are a stronger predictor of attitude similarity than individual attributes.

Despite the solid evidence showing the effect that being immersed in social structures has on social homogeneity, there is no full consensus about which relational process is more effective in explaining the variability of the elements. The first studies that applied SNA methods to explain social influence processes reveal that structural cohesion, being part of a dense network of direct contacts, is the main source of social homogeneity [21,22]. At the same time, other lines of research have proposed that role similarity, meaning that occupying equivalent positions within the same, or different, social networks, is the main determinant of social homogeneity [23]. Finally, a third mechanism is based on the effect of preferentially relating to actors with the same attributes (i.e., homophily) or to people with differential characteristics (i.e., heterophily) on social homogeneity [24]. Knowing what relational mechanism is more effective for explaining social homogeneity is relevant for several reasons. First, social homogeneity refers to a compositive feature of social groups; thus, by knowing what relational process best works to activate group homogeneity, it is possible to increase trust, commitment, and reciprocity, which are antecedents for group performance. Second, and in the opposite way, by identifying the relational process that triggers social homogeneity, we are also able to reduce it in some cases with the purpose of mitigating negative consequences such as group thinking and pressures towards group uniformity.

Given the implicit controversy that this implies, the main objective of this research is to determine which of the three relational processes described above (structural cohesion, role equivalence and homophily/heterophily) has the greatest impact on social homogeneity, understood as the similarity of attitudes and perceptions that individuals display toward the social context that surrounds them [9]. To achieve this objective, we will compare the effect of each mechanism in explaining the similarity of attitudes and perceptions in a sample of professionals implementing a psychosocial intervention program in Colombia. Although network theory suggests that the three relational processes that will be studied in this research contribute to social homogeneity, investigating the differential weight of each one of them can contribute to the literature on the antecedents that explain social homogeneity. The evidence derived from this work could help to explain formal and informal social cohesiveness and homogeneity of groups in organizational, community and other social entities. The following sections explain how structural cohesion, role equivalence, and homophily/heterophily operate, promoting social homogeneity.

### 1.1. Structural Cohesion

From a structural cohesion perspective, social homogeneity is explained by the fact that actors are connected to each other through multiple high-intensity relationships and tend to influence others through communication and prolonged interactions over time [23]. When a group of actors forms a cohesive subset, it usually emerges norms, obligations, and patterns of reciprocity that facilitate the development of similar attitudes, perceptions, and behaviors among group members [25]. These elements, in turn, contribute to the creation of systems of power and cross-pressures that influence group uniformity via social contagion.

As Burt and Janicik [26] point out, to understand the factors that precipitate the social homogeneity of members of cohesive groups, it is necessary to examine the structure of the entire network in which the cluster is embedded. This statement is relevant since one of the foundational principles of SNA assumes that macro-social phenomena determine micro-local processes and vice versa [27,28]. Therefore, it must be assumed that a certain degree of social homogeneity is explained by the impact of the properties of the social structure to which the cohesive subgroups in question belong.

Another essential element in understanding structural cohesion is that individuals can occupy different places within the network structure and that the position occupied determines access to information, resources, and ideas, and in general, to everything that flows through interactions [21,29]. The position that each actor occupies depends on multiple factors, including the number of contacts, the average distance that connects each participant with the rest of the network members, the intensity of the relationships, or the density of the entire network [30,31]. The dual notion of core-periphery helps to understand the concept of structural cohesion. Social networks can be fragmented into at least two structures called core and periphery [32]. Networks usually show a core-periphery structure. Actors occupying the center maintain multiple connections with each other and some links with peripheral actors, while peripheral actors are usually not connected to each other but maintain some contact with central actors. The central core of the network is considered a cohesive subset in which (a) the relational density is higher compared to the periphery; (b) the relationships, when weighted, are of greater intensity; (c) the average distance of the path connecting the actors in the center is lower; therefore (d) there is greater proximity between the actors in the center of the network compared to the actors located in the periphery [32]. Under these assumptions, the process of social contagion and, consequently, social homogeneity is intensified in the core of the networks compared to the periphery.

There are multiple parameters that serve to identify the actors that occupy central positions in social networks, such as degree and closeness centrality. There are also specific measures, such as the coreness indicator, that assess the degree to which each actor is close to the core of the network. Below, we present a simulated example to illustrate how it is possible to divide the structure of a social network into two distinct groups based on their proximity to the network core. Following the work of Borgatti and Everett [32], first, the categorical model is applied to differentiate the core and periphery structure. To do so, the actors assigned to each subgroup are identified, and then the continuous model is applied to calculate the value of the coreness parameter for each actor. In all the illustrations presented in this work, the calculation of the indicators and the clustering analysis are performed with the UCINET software [33]. Figure 1 illustrates the structural properties of a hypothetical network composed of 12 actors.

Figure 1 shows the new configuration of the adjacency matrix after running the core–periphery analysis, the density of relationships in the core, periphery, and core–periphery structure, and the ranking of the actors according to the score obtained in the coreness parameter. As can be seen, this analysis identifies the nodes occupying the core {1, 2, 3, 4, 5, 6, 7} and the periphery {8, 9, 10, 11, 12} of the network, and the coreness indicator reveals that the core actors have higher scores in this parameter. If we examine the density matrix, we can see that relational cohesion is four times higher in the core of the network (85.7%) compared to the periphery (20%).

This example illustrates the method that will be applied to show the mechanism that explains how structural cohesion affects social homogeneity. If structural cohesion does indeed affect the similarity of attitudes and perceptions, it should be possible to identify statistically significant differences in the evaluation of attitudes and perceptions between the core and periphery groups. Below, we examine the mechanism explaining the effect of role equivalence on social homogeneity.

### 1.2. Role Equivalence

The notion of equivalence is a classic research topic in SNA. Equivalence detection procedures aim to identify actors who have a similar way of relating to others in the same or different social networks. There are numerous types of equivalence depending on the criteria used to detect them. The most classic model, and the most restrictive one, is the structural equivalence model originally proposed by Lorrain and White [34]. This model considers that two actors are structurally equivalent only if they present the same relationships with the same actors. Therefore, structural equivalence can only occur in actors who are part of the same social network.

A less restrictive notion of equivalence is regular equivalence, a term coined by White and Reitz [35] to identify actors who present a similar pattern of connections with other actors in the same network or in different social structures. In this model, the criterion for fulfilling regular equivalence is relaxed, which makes it possible to identify subjects who, despite belonging to different networks, present a similar relational pattern. In this paper, we will employ a third type of equivalence, known as role equivalence [36], as a trigger for social homogeneity. Two actors are considered role equivalents when they are involved in identical patterns of relationships with other actors [23]. Under this assumption, two actors can exhibit role equivalence without requiring: (a) that they maintain a direct link with each other; (b) that they are connected to the same actors or; (c) that they are part of the same group. This notion of equivalence is less restrictive and better matched to real data compared to structural and regular equivalences [29,36,37].

Evidence suggests that role equivalence produces significant effects on social homogeneity [36]; however, the explanatory mechanism differs from that proposed for structural cohesion. In this type of equivalence, social homogeneity is determined by the fact that actors within the same equivalent class (of role in this case) inhabit similar relational contexts, resulting in similar responses to the same stimuli, which leads them to develop similar responses to the same stimuli which in turn promote them to develop similar attitudes and perceptions about their surrounding social environment [37]. Figure 2 presents a hypothetical example illustrating how it is possible to categorize actors based on the pattern of relationships they exhibit, even if they are not connected to each other and do not maintain contact with the same actors.

As can be seen in Figure 2, the networks of the two hospitals differ in size, morphology, and structural properties. Likewise, the degree centrality of the nodes of both networks varies notably, ranging from a minimum value of 5.882 to a maximum value of 35.294. However, the most relevant information is found in the table that identifies the subsets of regularly equivalent actors. The table shows how the actors that present the same pattern of relationships, measured by their degree of centrality, are included in the same subset, regardless of whether they belong to different hospitals. This analysis makes it possible to group the actors who present the same relational pattern independently of the network of which they form part. Thus, the role equivalence analysis groups the following subsets of actors {2, 11}, {1, 14, 18}, {4, 12, 16}, {8, 15, 17}, and {3, 5, 6, 7, 13} in the same class because they present the same value in degree centrality, even though they are part of different hospitals. While the subset formed by nodes 9 and 10 is structurally equivalent since both maintain identical connections with node 8 and both are part of the same hospital. The logic underlying this analysis is to identify actors who maintain the same relational pattern within the same or different social structures, with the purpose of determining whether showing the same pattern of relationships affects the similarity of actors’ attitudes and perceptions. The aim of this analytical strategy is to determine if role equivalence explains social homogeneity in terms of the similarity of attitudes and perceptions. This phenomenon is demonstrated if actors within the same role-equivalent class develop similar attitudes and perceptions compared to actors in other role-equivalent classes. In the following section, we will describe the role played by homophily and its antagonistic process, heterophily, in the generation of social homogeneity.

### 1.3. Homophily Versus Heterophily

Homophily describes the tendency of subjects to be in contact with other subjects with whom they perceive to have characteristics in common, while heterophily is the opposite process that consists of the preference of subjects to maintain relationships with people they perceive as different based on certain characteristics or attributes [38]. The presence of both relational patterns is observed in the selection of contacts that are part of personal networks [39,40], coworkers [41,42,43], or organizations with which companies enter into strategic alliances [44,45,46,47,48].

Different studies show that homophily leads to social homogeneity [49,50]. Homophily and heterophily occur through the activation of two psychosocial processes operating simultaneously: selection and influence [51,52,53]. Selection is involved in the decision to choose the people with whom to establish social bonds (e.g., people with the same professional profile in the case of homophily and people with different professional profiles in the case of heterophily). Influence acts by promoting convergence in the attitudes, perceptions, and behavior of individuals (e.g., developing similar attitudes toward the organization, perceiving the work unit climate similarly, or acting similarly in the face of adverse customer reaction). Since homophily leads to social homogeneity, the main trigger for homophily is the contagion process [54]. As Chancellor and colleagues [43] point out, to the extent that homophily and heterophily determine contact choice, this implies that both processes will lead to the grouping within the same cluster of those individuals showing a homophilic tendency, which will result in the formation of clusters of individuals with similar characteristics, or heterophilic, which will lead to the formation of clusters of individuals with differential characteristics.

Clustering within the same subgroups according to the relational pattern (homophilic versus heterophilic) progressively intensifies the relationships between the members of each cluster [40]. As the frequency and intensity of relationships increase, so does in-group cohesion, which usually leads to the emergence of clusters characterized by high density. Different proposals point to the fact that social influence processes are accentuated within cohesive subsets [55]. This occurs because group processes such as norms, conformism, and group pressure emerge in cohesive subsets leading to social homogeneity [8,9,19].

Other studies suggest that actors who exhibit role equivalence tend to demonstrate greater social homogeneity compared to those who are grouped within the same cohesive subset [37]. Assuming this premise, we consider that structural cohesion as well as homophily and heterophily, will produce less social homogeneity in perceptions and attitudes compared to playing equivalent roles. The mechanism leading to social homogeneity differs in the cases of structural cohesion and homophily/heterophily compared to role equivalence. In the first two assumptions, the contagion process occurs because actors in the same subgroup maintain high-intensity relationships with other group members, which implies that they are exposed to the same attitudes, beliefs, or ideas, whereas in the third assumption, people presenting role equivalence are connected in the same way to their relational environment, which facilitates them to emit similar responses to the same stimuli leading to social homogeneity [20,21,37]. Figure 3 describes the hypothetical network of a team of 10 psychologists and 10 social workers.

The lower left quadrant shows the egocentric networks of different professionals showing maximum levels of homophily and heterophily (nodes 5 and 18, respectively) and moderate homophily and heterophily (nodes 2 and 11, respectively). The parameter to measure homophily is the E-i index, whose value ranges from 1, denoting pure heterophily, to −1, reflecting pure homophily [56]. From the values of the E-i index, it is possible to group the actors into clusters according to their level of homophily or heterophily so that it is possible to identify clusters formed by nodes that show a pattern of relationships based on homophily and heterophily. According to the background information reviewed, it is expected that clusters whose actors maintain relationships based on homophily will show greater social homogeneity compared to clusters where no clear homophilic tendency is observed. The following section presents the overall goal of this research and formulates three working hypotheses in response to that goal.

### 1.4. Research Objective

The general purpose of this study is to determine which of the relational processes explored (structural cohesion, role equivalence, and homophily/heterophily) produces the greatest effect on social homogeneity in a group of professionals implementing a psychosocial intervention program. To achieve this overall goal, three working hypotheses need to be formulated:

**Hypothesis 1.** *Actors grouped within the core structure of the evaluated networks will present greater social homogeneity compared to actors located in the periphery, with the difference between the core and the periphery being statistically significant*.

**Hypothesis 2.** *Actors grouped within the same equivalent role class will present a distinguished level of social homogeneity compared to actors located within another class, with the difference between the two groups being statistically significant*.

**Hypothesis 3.** *Actors grouped within the same cluster identified according to the E-i index in the three types of relationships explored will present a level of social homogeneity distinguishing them from that of actors located within another cluster, with the difference between the two groups being statistically significant*.

## 2. Materials and Methods

### 2.1. Context of the Study

This research was developed within the framework of a psychosocial intervention program aimed at victims of the armed conflict in Colombia. The program is implemented throughout the country, although its management is decentralized in different territories. In the three regions where the evaluated professionals provide care, the program serves an annual average of 25,000 users. The program adopts a systemic and multilevel intervention approach, providing individual, family, and community care to victims of the armed conflict included in the Registro Único de Víctimas (Unique Registry of Victims: https://www.unidadvictimas.gov.co/es/registro-unico-de-victimas-ruv/37394 (accessed on 23 July 2022)). A precise description of the intervention program is provided in previous works [57,58].

### 2.2. Participants

The research involved professionals (*N* = 110) who implement a psychosocial intervention program in three regions of Northwest Colombia. In the region of Córdoba, 49 professionals participated (44.5%), in Bolívar 34 (30.9%) and in Atlántico 27 (24.5%). Most of the participants were women (*n* = 87; 92%) who, on average, have been implementing the program for 16.9 months (*SD* = 14.7). The professionals implementing the program were psychologists (*n* = 50; 45.5%), social workers (*n* = 35; 31.8%), and community advocates and administration and services personnel (*n* = 25; 22.8%). To gain access to the participants, prior contact was established with those responsible for the program in each of the regions. The program coordination office provided the research team with a list of contact details of the implementers who were interviewed between September 2017 and August 2018. The only criterion for inclusion was to have been working as a program implementer for more than one month.

### 2.3. Variable to Assess Social Homogeneity

The construct known as Psychological Sense of Community (PSC) was used to assess social homogeneity. This decision was made because the PSC includes aspects of a perceptual and attitudinal nature that are relevant descriptors for assessing social homogeneity in specific contexts such as the one described in this research [9]. To assess PSC, we started with McMillan’s theoretical model, which was later operationalized in four dimensions (needs fulfillment, belongingness, influence, and shared emotional connection) by McMillan and Chavis [59]. In the original work, McMillan [60] defined PSC as: “a feeling that members have of belonging, a feeling that members matter to one another and to the group, and a shared faith that members’ needs will be met through their commitment to be together”. This construct assesses the attitudes of the group members toward the context that surrounds them. In this research, PSC assesses the attitudes of professionals toward (a) the work they carry out as implementers, (b) the team of professionals who implement the program, and (c) the psychosocial intervention program for which they provide services. The theoretical model proposed by McMillan and Chavis [59] was conceived to evaluate PSC with respect to geographically delimited communities (e.g., neighborhoods) and with respect to relational communities. PSC has been recurrently assessed in organizations making it a suitable construct to assess social homogeneity in the context of the study [61,62,63].

The second version of the Sense of Community Index (SCI-2) [64] was used to evaluate PSC. The SCI-2 consists of 24 items evaluated on a four-point Likert scale (1 = do not agree at all and 4 = completely agree). Each dimension is assessed by six items. The dimension of needs fulfillment examines the degree to which professionals consider that by participating in the intervention program, they contribute to satisfying the needs of other team members. While participating in the team, they themselves are able to fulfill different types of needs. An example of an item in this dimension is the following: “*The members of my work team and I value the same things*”. The belonging dimension assesses the degree to which team members consider that they are part of a higher-order social structure, in this case, the group of professionals implementing the program in each region. An example of an item in this dimension is the following: “*Being a member of the team implementing the program is a part of my identity*”. The influence dimension describes the degree to which team members are aware that by being part of the team, they can influence and be influenced by other professionals implementing the program. An example of an item in this dimension is the following: “*I have influence over the team of professionals implementing the program*”. Finally, the shared emotional connection dimension assesses the extent to which the professionals implementing the program have shared symbols and codes that allow differentiating the members of the in-group from those of the out-group. An example of an item in this dimension is the following: “*I hope to be part of this work team for a long time*”. The complete scale has excellent psychometric properties (α = 0.88), while the reliability of the subscales ranges between 0.70 and 0.73. To determine whether the dimensions of the PSC follow a normal distribution, the Kolmogorov–Smirnov test was performed. This analysis revealed that none of the dimensions of the PSC shows a normal distribution in the sample. In addition, the same test was performed to determine whether there are significant differences in the four dimensions of the PSC according to the professional profile of the participants, and no differences were found.

### 2.4. Measures for Assessing Structural Cohesion, Role Equivalence and Homophily/Heterophily

To test the working hypotheses, different methodological strategies were developed. To build networks among professionals implementing the program, a socio-centric questionnaire was designed and applied to program implementers in each region [57,58]. The relationships assessed were recognition among professionals, information exchange, and user referral. Previous studies suggest that these relationships are especially relevant in the context of intervention programs implemented by interprofessional teams [65,66]. The relational data were transferred to binary adjacency matrices processed with UCINET software [33], which was used to calculate centrality measures, the core–periphery structure of the three types of networks in each region, and the level of homophily/heterophily of each actor using as a reference attribute the professional profile of the implementers (psychologists, social workers, and community promoters and support staff). The output of the relational analysis was three networks in each region (nine networks in total), one for each type of relationship. Figure 4 shows the recognition, information exchange, and user referral networks in the three regions. Letters A, B, and C identify the networks of the teams implementing the program in the regions of Atlántico, Bolívar, and Córdoba, respectively. The size of the node represents the length of time each professional has been working in the program, while the color of the node indicates the professional profile (black = psychologists, red = social workers, and blue = community promoter/administration and services personnel).

### 2.5. Analysis Strategy

Three different procedures were developed to group implementers according to structural cohesion, role equivalence, and degree of homophily/heterophily according to the professional profile. To test the first hypothesis, we first applied the core–periphery analysis following the categorical model proposed by Borgatti and Everett [32] on the recognition, information exchange, and user referral networks in each region. This analysis assigns each professional to the core or periphery of the network, depending on his or her proximity to each structure. By definition, structural cohesion is higher in the core compared to the periphery (see Figure 1). The categorical algorithm identified 70 professionals (63.6%) in the core and 40 (36.3%) in the periphery recognition network; in the information exchange network, it identified 45 professionals (40.9%) in the core and 65 (59.1%) in the periphery; in the user referral network, it identified 35 professionals (31.8%) in the core and 75 (68.2%) in the periphery.

To test the second hypothesis, the degree of centrality of each professional in the three networks evaluated in each region was calculated. Degree centrality is an indicator composed of outdegree and indegree nominations that each actor submits and receives in each of the networks examined. To classify professionals according to the degree of centrality they present in the different networks, cluster analysis was applied using the k-means method. For this purpose, the degree centrality of each actor in the recognition, information exchange, and user referral networks was used as the clustering variable. The analysis identified, after four iterations, an optimal solution of two clusters, with the minimum distance between the initial centers of each cluster being 1.091. The first cluster was made up of 61 professionals (62.72%) and was characterized by a moderate level of degree centrality in the recognition network (0.41) and a low level of centrality in the information exchange (0.10) and user referral (0.05) networks. The second cluster was made up of 49 professionals (37.27%) and was distinguished by showing a high level of degree centrality in the recognition network (0.75) and medium–low levels of centrality in the information exchange (0.25) and user referral networks (0.12).

To test the third hypothesis, we first calculated the E-i index [56] of each professional in the three networks evaluated, taking as a reference attribute the professional profile of each participant. To classify the professionals according to the values of the E-i index in the three types of relationships explored, cluster analysis was performed using the k-means method, using the E-i index as clustering variables in this case according to the professional profile in the recognition, information exchange, and user referral networks. The analysis converged to an optimal two-cluster solution after six iterations, with the minimum distance between the initial centers of each cluster being 3.464. The first cluster was made up of 49 actors (37.27%) and was characterized by a slight homophilic tendency in the recognition network (E-i = −0.07) and a moderate tendency in the information exchange (E-i = −0.23) and user referral (E-i = −0.24) networks. The second cluster was made up of 61 professionals (62.72%) and presented medium–high levels of heterophily as a function of professional profile in the recognition (E-i = 0.49), information exchange (E-i = 0.51), and user referral (E-i = 0.63) networks. To test the three working hypotheses, different non-parametric tests were performed because the variable selected to evaluate the level of social homogeneity (PSC) does not have a normal distribution in the sample. Table 1 shows the descriptive statistics of the variables analyzed.

## 3. Results

### 3.1. Hypothesis 1

To test the first hypothesis, three non-parametric tests (Kruskal–Wallis H) were developed using the core/periphery membership in the recognition, information exchange, and user referral networks as clustering variables. Table 2 shows the results of the three non-parametric tests carried out, using the Monte Carlo significance criterion with a 99% confidence interval, and highlighting in bold the significant values of the Kruskal–Wallis H index and the *p*-values.

As can be seen in Table 2, statistically significant differences can only be observed between the core and periphery structure in the dimension of belonging, both in the network of recognition among professionals and in the user referral network. In both networks, the average rank of the belonging dimension was notably higher in the core of the network compared to the periphery. These data suggest that the actors located in the core structure of the networks mentioned experiencing a greater sense of belonging compared to the actors located in the periphery. However, only one of the dimensions used to measure social homogeneity, in this case belonging, showed statistically significant differences between professionals located in the core and in the periphery of the networks evaluated. Moreover, this difference occurred in only two of the three networks examined. Therefore, the hypothesis of structural cohesion to explain social homogeneity was only partially supported since, to be fully satisfied, the differentiation between core and periphery structure may have been expected to occur in all four dimensions of the PSC and in the three types of networks studied.

### 3.2. Hypothesis 2

The second hypothesis proposed that role equivalence, namely, that actors are grouped within the same cluster by presenting a similar pattern of connections based on degree centrality in the three networks assessed constitutes an explanatory factor for social homogeneity. Two clusters were identified based on degree centrality. The first cluster included professionals (*n* = 61) who showed a moderate degree of centrality in the three networks explored, while professionals grouped in the second cluster (*n* = 49) exhibited a medium–high degree of centrality in all three networks. As can be seen in Table 3, in the four dimensions of the PSC, the average rank was higher in the second cluster compared to the first. Therefore, the actors that presented a similar connectivity pattern in the different networks developed similar attitudes toward the work they carried out in the program, the team of professionals implementing the program, and the program itself. Considering that the differences were statistically significant between both groups in all the dimensions of the variable used to measure social homogeneity, the second study hypothesis was fully supported.

### 3.3. Hypothesis 3

The third hypothesis proposed that homophily is an explanatory factor for the similarity of perceptions and attitudes experienced by the evaluated professionals. To test this hypothesis, each participant was assigned to a cluster, using as a grouping variable the E-i index in the recognition, information exchange, and user referral networks. Table 4 shows the results of the non-parametric analysis used to test the last hypothesis.

The non-parametric tests demonstrated that there are statistically significant differences between the two clusters in terms of the E-i index in the three networks examined for the dimensions of belonging and shared emotional connection. In both dimensions, differences were observed between the first cluster, characterized by a low level of homophily, and the second cluster, with a medium–high level of heterophily. These results offer partial support for the third hypothesis given that, although significant differences were observed between the two clusters according to the dimensions of the PSC, these differences were only observed in two of the four dimensions of the variable used to measure social homogeneity.

Another outstanding result is that in the first cluster (*n* = 49), the levels of all PSC dimensions were substantially lower compared to those observed in the second cluster. This finding will be commented on in more detail in the discussion since it seems to contradict the results obtained in previous studies.

In order to achieve the overall goal of this research, it is necessary to examine the results derived from contrasting the three hypotheses in an aggregate form. Figure 1 presents the values of the H index that reports the differences in the four dimensions of the PSC between the clusters detected by the structural cohesion (core vs. periphery), role equivalence (cluster formed by actors with moderate levels of degree centrality vs. cluster formed by actors with high levels of degree centrality), and homophily/heterophily (cluster formed by actors with moderate homophily vs. cluster formed by actors with noticeable heterophily) approaches.

Figure 5 provides a visual representation of the values of the H index that are significantly higher in the case of role equivalence compared to the values identified in the difference between core and periphery, illustrating structural cohesion (in the recognition, information exchange and user referral networks), and to the values observed in the case of clusters identified according to the level of homophily/heterophily. Moreover, this phenomenon is observable in all four dimensions of the PSC. Therefore, role equivalence, that is, the clustering of participants in terms of presenting a similar pattern of relationships in the different networks analyzed, constitutes the main mechanism that explains social homogeneity. The following section discusses the findings of this research.

## 4. Discussion

Previous studies suggest that social homogeneity, understood as the similarity of attitudes and perceptions that individuals develop in relation to the social context that surrounds them, is largely explained by relational processes such as influence [5,6,19,67,68], contagion [21,26,54,69,70,71], or homophily [38,42,51,52]. However, there is no explicit agreement on which structural mechanism has the greatest power to explain social homogeneity.

Previous studies compared the impact of structural cohesion and different types of equivalence (structural, regular, and role equivalence) on social homogeneity [22,23,31], but the findings cannot be considered conclusive. For this reason, this research arose to overcome this gap in the literature by determining which of the three relationship processes examined had the greatest power to explain the social homogeneity concerning the attitudes and perceptions developed by a group of professionals implementing a psychosocial intervention program. The results of this study were enlightening since they revealed that role equivalence, understood as the degree to which two actors are considered equivalent because they maintain a similar pattern of relationships, was the main factor explaining social homogeneity in the sample under study.

To examine what factors promote role-equivalent actors to develop similar attitudes, it was necessary to understand the relational context in which the interactions between the participants in this study occur. In this regard, Mizruchi [23] proposed that role equivalence measured through the prominence (i.e., degree centrality) of an actor within one or more social networks acts as a powerful trigger of social homogeneity when networks are small and highly stratified (p. 276). If we observe Figure 4, containing the description of the cohesion parameters of the evaluated networks, we can see that all the networks were relatively small (n < 50 actors) and presented a density lower than 50% with the exception of the recognition network in the Atlántico and Córdoba regions, and the information exchange network in the Atlántico region, and show a low transitivity that in most of the networks does not exceed 40%. This aspect may amplify the effect of role equivalence on social homogeneity.

The social forces that emerge in networks determine individuals’ perceptions of the socio-professional context surrounding the performance of their functions [72]. This phenomenon means that professionals, when exposed to the same relational environment, end up emitting similar responses to the same environmental stimuli, which constitutes a source of social homogeneity. In this research, we applied the concept of role equivalence, understood as a flexible modality of regular equivalence [29] that allowed us to evaluate the similarity in the pattern of relationships maintained by a set of actors who were part of different social structures. The participants in this research implemented the same intervention program but in different regions; this fact made it advisable to apply a more relaxed criterion of equivalence than the orthodox notion of structural equivalence proposed by Lorrain and White [34]. On the other hand, the operational definition of role equivalence was carried out using the degree centrality of professionals in different types of relationships as suggested by previous studies [37], which enabled the acquisition of a complete view of the social system constituted by the networks of professionals implementing the program.

The results allow us to assert that, at least in the context of this research, role equivalence is the most powerful mechanism to explain the similarity of attitudes and perceptions experienced by professionals toward their own work, toward the team of professionals implementing the program, and toward the program itself. If we compare the role equivalence hypothesis with the structural cohesion hypothesis, we can draw two interesting lessons. Firstly, in this case, being part of the central core of the network seems to not cause the expected contagion effect among the actors occupying the core of the social networks [71]. The moderate level of density that characterizes most of the evaluated networks possibly minimizes the degree of contagion derived from exposure to direct contacts in multiple types of relationships. Secondly, the effect on social homogeneity of occupying the core and periphery structure varies substantially depending on the type of PSC dimension assessed. The only dimension in which statistically significant differences were observed between professionals occupying the core and the periphery of the network was belonging. This dimension is the most relevant of the PSC because it describes the degree to which professionals perceive that they are part of a social structure of a higher order, differentiated from other social systems [59]. A key aspect lies in the need to encourage interpersonal relationships among implementers to create a sense of belonging to the team and to the program itself with the purpose of promoting the involvement of professionals in the implementation process [70].

Finally, an outstanding result referring to the third working hypothesis contrary to the evidence reported in previous studies was that the levels of PSC identified in the second cluster characterized by high levels of heterophily were significantly higher than those observed in the cluster characterized by a moderate level of homophily. The literature suggests that homophily is a precursor of social homogeneity [38,49,50], so one would expect to identify higher levels of PSC in the first cluster compared to the second, which was not the case. This fact may be due to several factors that should be taken into consideration to establish the configuration of the teams that implement this type of program. First of all, if the program is implemented by interprofessional teams, it is probable that the team configuration itself encourages heterophilic relationships, something that would reinforce the identification of higher levels of PSC in the heterophilic cluster compared to the homophilic cluster. Furthermore, in this context, it may be more satisfying for professionals to interact with professionals of different profiles since they can exchange intervention practices and experiences from another professional perspective, which may translate into higher levels of effectiveness as perceived in the performance of their functions. Finally, homophilic relationships can lead to a certain level of exhaustion due to the similarity of attributes, in this case regarding professional profile. This phenomenon may contribute to the overlapping of interests and objectives and the saturation of available information, factors that may lead to a lack of effectiveness in the execution of tasks, disaffection toward the work, toward the team of implementers, and toward the intervention program implemented [73].

### Limitations

The current study presents a series of shortcomings that deserve to be pointed out. The conclusions of this research are derived from analyzing a small and intentional sample so the results are hardly applicable to other organizational environments without proper contextualization. On the other hand, the organizational dynamics that guide the program’s implementation process may be influencing the establishment of relationships between professionals and, consequently, the structure and composition of the networks that have been evaluated to test the proposed hypotheses. Another factor to take into consideration is that the professionals implement the program in different regions, so it is possible that the operational patterns that determine the execution of the program in each region (i.e., coordination system, leadership styles, staff turnover, etc.) affect the relational processes examined in this work. Finally, the data do not follow a normal distribution in the sample, which has led to the inability to apply more powerful statistical tests to analyze the hypotheses under study.

## 5. Conclusions

SNA offers researchers and practitioners a wide range of theories and methods to explain the grouping dynamics and compositive variables. This work focuses on a relational mechanism based on the principle of equivalence, which produces a notable impact on the level of social homogeneity in attitudes in a group of professionals implementing a psychosocial intervention program for war victims in Colombia. This finding can help activate processes of social influence within the group aimed at modifying positive attitudes towards the work they perform and the users of the service they provide. For this reason, activating relational processes oriented to increase social homogeneity could be a pragmatic option to reinforce psychosocial factors such sense of community, work engagement, and prosocial behaviors, which are essential for individual and group performance.

The main contribution of this paper is to show that role equivalence is an important relational mechanism that determines social homogeneity in the organizational context examined. The current evidence indicates that the study of the factors leading to social homogeneity must be carried out in the relational context in which individuals develop. Therefore, the object of study to understand the mechanisms leading to homogeneity should be social networks rather than isolated individuals [20]. From this perspective, the results suggest that the relational pattern assessed by the degree centrality of actors in different networks is the relationship phenomenon that best explains social homogeneity compared to the positioning in the core or the periphery of the networks (structural cohesion) or to the preference to maintain contacts based on the perception of similarity (homophily) or dissimilarity (heterophily).

## Figures and Tables

**Figure 1 ijerph-19-14471-f001:**
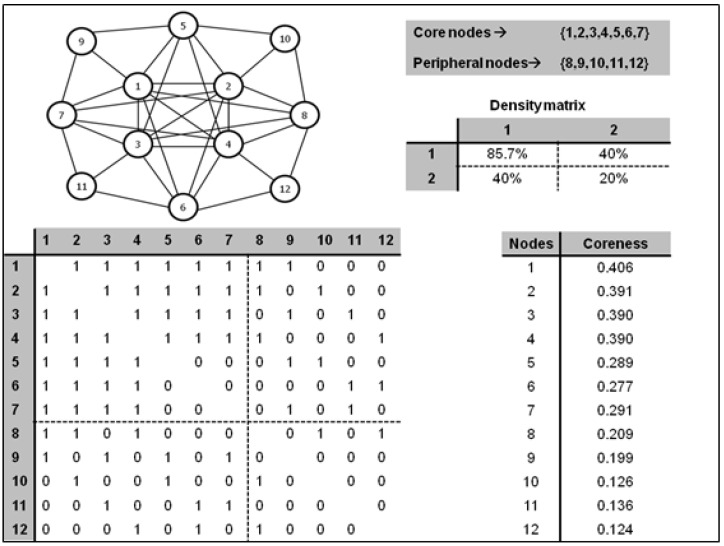
Visual and analytical illustration of structural cohesion using core–periphery analysis.

**Figure 2 ijerph-19-14471-f002:**
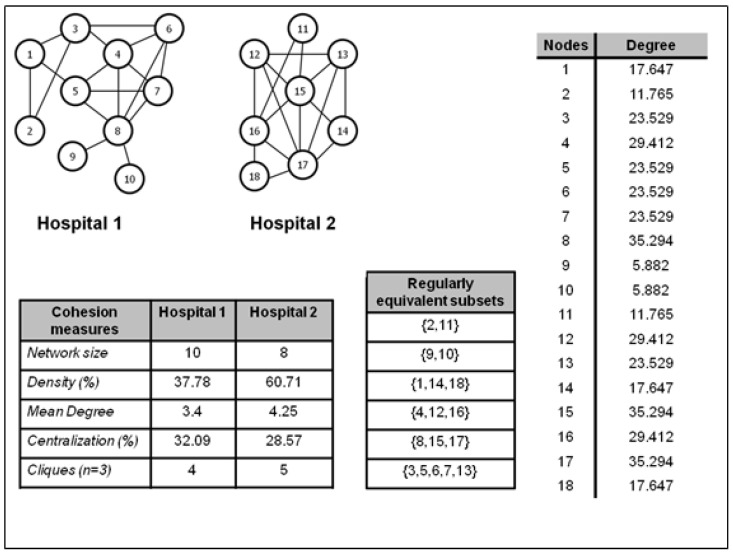
Illustration of role equivalence identifying the regularly equivalent subsets of actors, the cohesion measures of the networks of the two hypothetical hospitals, and the degree centrality of all nodes.

**Figure 3 ijerph-19-14471-f003:**
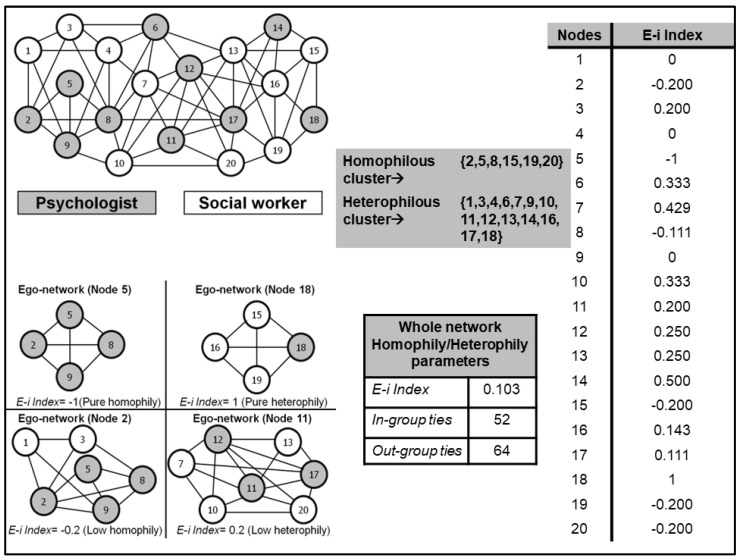
Illustration of homophily/heterophily according to professional profile identifying two clusters using the E-i index as a grouping variable. The lower left quadrant shows the egocentric networks of four professionals depending on the degree of homophily/heterophily of their egocentric network.

**Figure 4 ijerph-19-14471-f004:**
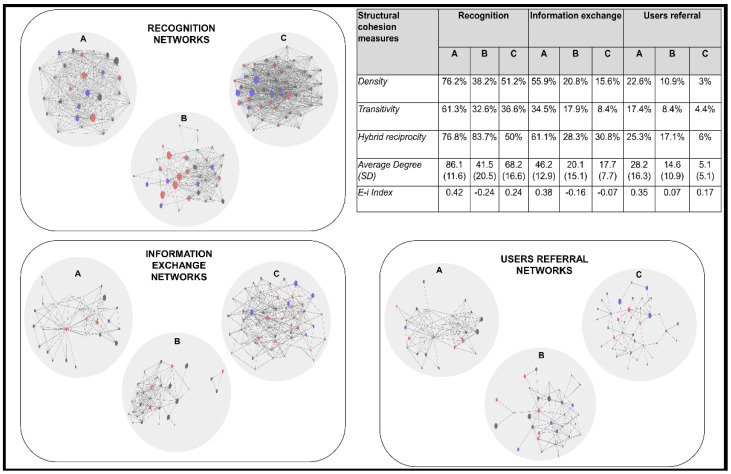
Recognition, information exchange and user referral networks among program implementers in the region of Atlántico, Bolívar, and Córdoba and description of the cohesion parameters of each network.

**Figure 5 ijerph-19-14471-f005:**
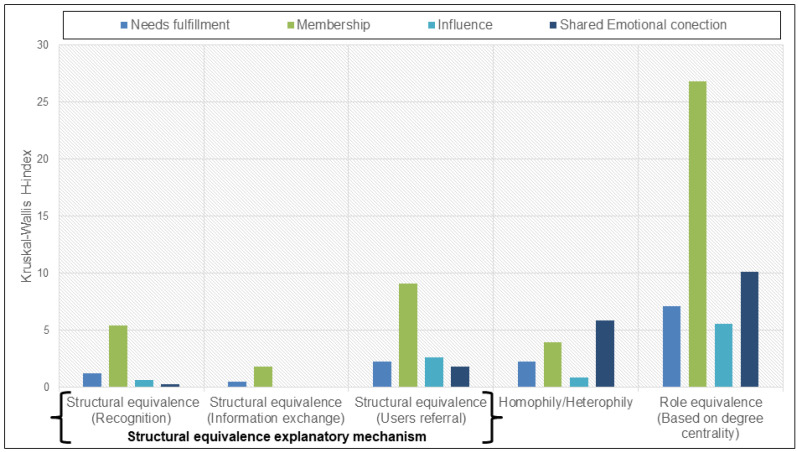
Kruskal–Wallis H-index values in the PSC dimensions in the structural cohesion, homophily/heterophily and role equivalence approaches.

**Table 1 ijerph-19-14471-t001:** Descriptive statistics of the variables analyzed (*N* = 110).

Variables	Min.	Max.	M	SD
Needs fulfillment (PSC)	1.5	4	3.26	0.44
Belonging (PSC)	1.67	4	3.38	0.42
Influence (PSC)	2	4	3.20	0.45
Shared emotional connection (PSC)	1.67	4	3.29	0.49
Degree centrality (recognition network)	0.08	1.02	0.56	0.22
Degree centrality (information exchange network)	0.03	0.61	0.16	0.11
Degree centrality (user referral network)	0.02	0.29	0.08	0.06
E-i Index (recognition network)	−1	1	0.24	0.42
E-i Index (information exchange network)	−1	1	0.17	0.51
E-i Index (user referral network)	−1	1	0.24	0.59

Note: PSC = Psychological Sense of Community.

**Table 2 ijerph-19-14471-t002:** Kruskal–Wallis H test applying the Monte Carlo significance criterion with a 99% confidence interval.

Contrast Variables	Recognition Network					Information Exchange Network					Users Referral Network				
	Struct.	AR	H	*p*	CI (99%)	Struct.	AR	H	*p*	CI (99%)	Struct.	AR	H	*p*	CI (99%)
NF	Core	52.51	1.231	0.271	0.260–0.283	Core	57.49	0.466	0.501	0.487–0.513	Core	61.69	2.254	0.135	0.126–0.144
	Per.	59.46				Per.	53.32				Per.	51.97			
Bel.	Core	60.7	**5.422**	**0.020**	0.016–0.023	Core	60.39	1.820	0.175	0.165–0.184	Core	68.83	**9.116**	**0.002**	0.001–0.003
	Per.	46.01				Per.	52.12				Per.	49.28			
Infl.	Core	53.22	0.628	0.438	0.425–0.450	Core	55.25	0.005	0.947	0.941–0.953	Core	62.04	2.594	0.111	0.103–0.119
	Per.	58.19				Per.	54.83				Per.	51.67			
SEC	Core	56.09	0.237	0.630	0.617–0.642	Core	55.27	0.006	0.943	0.937–0.949	Core	60.91	1.828	0.166	0.156–0.175
	Per.	53.04				Per.	54.81				Per.	52.20			

Notes: NF = Needs fulfillment; Bel. = Belonging; Infl. = Influence; SEC = Shared emotional connection; AR = Average range; Struct. = Structure; Per. = Periphery. Bold numbers identify statistically significant differences.

**Table 3 ijerph-19-14471-t003:** Non-parametric tests for assessing the role equivalence hypothesis.

Contrast Variables	Cluster	AR	H	*p*	CI (99%)
NF	1	47.91	**7.094**	**0.007**	0.004–0.009
	2	64.01			
Bel.	1	41.51	**26.800**	**0.000**	0.000–0.000
	2	72.92			
Infl.	1	48.61	**5.532**	**0.018**	0.015–0.022
	2	62.83			
SEC	1	46.51	**10.125**	**0.001**	0.000–0.000
	2	65.79			

Notes: NF = Needs fulfillment; Bel. = Belonging; Infl. = Influence; SEC = Shared emotional connection; AR = Average range. Bold numbers identify statistically significant differences.

**Table 4 ijerph-19-14471-t004:** Non-parametric tests for assessing the homophily hypothesis.

Contrast Variables	Cluster	AR	H	*p*	CI (99%)
FN	1	49.94	2.239	0.134	0.125–0.143
	2	58.98			
Bel.	1	48.83	**3.934**	**0.046**	0.041–0.051
	2	60.86			
Infl.	1	51.99	0.818	0.369	0.356–0.381
	2	57.46			
SEC	1	46.93	**5.878**	**0.015**	0.012–0.018
	2	61.59			

Notes: NF = Needs fulfillment; Bel. = Belonging; Infl. = Influence; SEC = Shared emotional connection; AR = Average range. Bold numbers identify statistically significant differences.

## Data Availability

The dataset of this research corresponding to the Department of Cordoba is available in the OPEN ICPS repository: https://www.openicpsr.org/openicpsr/project/115230/version/V1/view;jsessionid=F709F652E6C5EE7B29779B9C3CF319B0 (accessed on 13 April 2022). Ramos-Vidal, Ignacio. Structural evaluation of the implementation of the Psychosocial Intervention Program for Victims of the armed conflict in Colombia (PAPSIVI). Ann Arbor, MI: Inter-university Consortium for Political and Social Research [distributor], 30 October 2019. https://doi.org/10.3886/E115230V1.

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
