# Peer review of "Structural Cohesion, Role Equivalence, or Homophily: Which Process Best Explains Social Homogeneity?"

_ijerph, 2022, doi:10.3390/ijerph192114471_

Round 1

Reviewer 1 Report

1.         The introductory part contains clarifications of the main concepts, presentation of famous studies related to explaining collective behaviour. These are well related to the issue of social cohesion. However, it should be made as clear as possible what the practical importance of knowing the factors that determine social cohesion is. Where does the author think the knowledge methods he proposes will be used and by whom?

2.         The author identifies three relational processes that determine the manifestation of social cohesion. He then sets out to identify which of the three relational processes most determines social cohesion. Some issues emerge that should be clarified:

a.         Why a hierarchy of influence is indicated when we know by theory that a comprehensive explanation of social cohesion must take into account all the factors that determine the phenomenon (structural cohesion, role equivalence and homophily)?

b.         To what extent can the results of this study be generalized to communities other than those considered in Colombia?

c.         To what extent can the results be generalised to social phenomena other than cohesion among victims of armed conflict?

These clarifications would be appropriate in sub-chapter 2.1 "context of the study".

 3.         In relation to Figure 5, page 13, the picture related to structural cohesion is divided into three dimensions, which may induce a misperception of the degree of influence on social cohesion. I therefore suggest that the three dimensions of structural cohesion should be combined into one dimension. In this way the graph will better express the relationship between the relational processes explored (structural cohesion, role equivalence and homophily). The forms of manifestation of structural cohesion (recognition, information exchange, users referral) can be included in a separate table.

  4.         In the conclusion section I suggest a broader discussion of the benefits of social network analysis. I believe that from the data obtained in the study some strategies could be suggested to increase social cohesion through ways of influencing currents of opinion based on relationships within social networks.

Author Response

Dear reviewers,

Thank you so much for your time and efforts reviewing the manuscript. Your comments are essential to improve the overall merit of the article. In these lines I will try to response to the suggestions made by reviewers. Thank you again for your contributions.

Response to reviewer#1

Comment 1: The introductory part contains clarifications of the main concepts, presentation of famous studies related to explaining collective behaviour. These are well related to the issue of social cohesion. However, it should be made as clear as possible what the practical importance of knowing the factors that determine social cohesion is. Where does the author think the knowledge methods he proposes will be used and by whom?

Response: (Lines 85-93): To know what relational mechanism is more effective for explaining social cohesion is relevant for several reasons. First, social cohesion refers to a core structural feature of social groups, thus, knowing what relational process best work activating group cohesiveness, it is possible to increase trust, commitment and reciprocity which are ante-cedents for group performance. Second, and in an opposite way, identifying the relational process that trigger social cohesion, we also can be able to reduce social cohesion in order to mitigate negative consequences of excessive social cohesion such for example group thinking and pressures towards group uniformity.

Comment 2a: The author identifies three relational processes that determine the manifestation of social cohesion. He then sets out to identify which of the three relational processes most determines social cohesion. Some issues emerge that should be clarified:

  1. Why a hierarchy of influence is indicated when we know by theory that a comprehensive explanation of social cohesion must take into account all the factors that determine the phenomenon (structural cohesion, role equivalence and homophily)?

Response: I appreciate and agree with this comment. Social cohesion is explained in certain (and usually overlapping degree) by the three relational mechanisms mentioned. However, in some cases, for example when the group is too homogeneous, it is not possible to applied relational explanations as homophily. This text is included (Lines 101-104): Although network theory suggests that the three relational processes that will be studied in this research contribute to social cohesion, investigating the differential weight of each one of them can suppose a contribution to the literature on the antecedents that explain social cohesion.

Comment 2b: To what extent can the results of this study be generalized to communities other than those considered in Colombia?

Response: (Lines 104-106): The evidence derived from this work could help to explain formal and informal social cohesiveness of groups in organizational, community and other social entities.

Comment 2c: To what extent can the results be generalised to social phenomena other than cohesion among victims of armed conflict?

Response: The previous response can also respond to comment 2c. Also the cohesion evaluated in this work does not evaluate social cohesion among war victims but it is evaluated cohesion in a group of program implementers who provide psychosocial attention to war victims.

Comment 3: In relation to Figure 5, page 13, the picture related to structural cohesion is divided into three dimensions, which may induce a misperception of the degree of influence on social cohesion. I therefore suggest that the three dimensions of structural cohesion should be combined into one dimension. In this way the graph will better express the relationship between the relational processes explored (structural cohesion, role equivalence and homophily). The forms of manifestation of structural cohesion (recognition, information exchange, users referral) can be included in a separate table.

Response: I especially appreciate this comment, and in fact, it is a matter that I was able to discuss with close colleagues during the development of the work. However, I chose to indicate differentially the results of structural cohesion because in the three networks, the impact of cohesion was substantially different on the dimensions of sense of community. However, I have introduced a modification to the graph to make it easier for the reader to differentiate between the three relational mechanisms. Thanks again for your observation. (See figure 5, page 14).

Comment 4: In the conclusion section I suggest a broader discussion of the benefits of social network analysis. I believe that from the data obtained in the study some strategies could be suggested to increase social cohesion through ways of influencing currents of opinion based on relationships within social networks.

Response: (Lines 637-647): SNA offers to researchers and practitioners a wide range of theories and methods to explain the grouping dynamics and inner workings of social structures. This work focuses on a relational mechanism based on the principle of equivalence, which pro-duces a notable impact on the level of social cohesion of a group of professionals that is part of a broader organizational structure. This finding can help activate processes of social influence within the group aimed at modifying positive attitudes towards the work they perform and towards the users of the service they provide. For this reason, activating relational processes oriented to increase social cohesion could be a pragmatic option to reinforce psychosocial factors such sense of community and prosocial behavior which are essential for individual and group performance.

Thank you so much to reviewer 1 for their helpful comments.

Reviewer 2 Report

Dear authors, your article aroused my keen interest. You have proposed original methodological methods for studying a scientific problem.
However, I want to make suggestions to improve the content of the article.
1. You need to discuss with the editor the possibility of using a non-strict title of your scientific article. Although I have seen similar titles of articles in other magazines.
2. It is necessary to remove the word "Colombia" from the keywords to the article. It is not a scientific concept.
3. You start the narrative of the article with the concepts of attitude and social perception, but this is not the main thing. It is necessary to start with the problem of identifying indicators of social homogeneity. And after that, go to the concepts of attitude and perception.
4. I have a question. Is the trend you have identified a cultural phenomenon? Colombia is a country of collectivist values, isn't it?
5. The respondents' profession is socially oriented. Can this affect their equivalence of roles?
6. In the list of references, out of 73 literary sources, only 11 are dated for the last 5 years.
In general, your research is very interesting and non-standard

Author Response

Dear reviewer 2:

Response to reviewer#2

Comment 1: You need to discuss with the editor the possibility of using a non-strict title of your scientific article. Although I have seen similar titles of articles in other magazines.

Response: I agree in some extent with your comment. I propose this alternative title: “Unraveling the relational mechanisms that contribute to explain social cohesion”

Comment 2: It is necessary to remove the word "Colombia" from the keywords to the article. It is not a scientific concept.

Response: Done! (See line 24).

Comment 3: You start the narrative of the article with the concepts of attitude and social perception, but this is not the main thing. It is necessary to start with the problem of identifying indicators of social homogeneity. And after that, go to the concepts of attitude and perception.

Response: (see lines 28-31): Understand the processes which conditioning the structure and dynamics of groups it is a core question for social scientists. Empirical evidence highlights that many psychosocial phenomena that determine individual behavior are, at least partially, explained by the relational context in which individuals are immersed.

Comment 4: I have a question. Is the trend you have identified a cultural phenomenon? Colombia is a country of collectivist values, isn't it?

Response: I appreciate the question formulated by reviewer 2, because it takes into account the influence of structural variables of a country (in this case the collectivist orientation of culture) on phenomena at the micro or meso social level. I agree with the reviewer that Colombia is a country that stands out for its collectivist orientation and this probably affects the way of relating to the implementers who have participated in this study. However, with the data collected it is not possible to examine the effect that the culture variable produces on the dynamics of the evaluated team. I really appreciate your input.

Comment 5: The respondents' profession is socially oriented. Can this affect their equivalence of roles?

Response: Probably, the own nature of this kind of profession may determine the way to form and maintain relationships with other colleagues in a given setting. However, literature suggest that direct effects created by organizational culture, norms, hierarchy, leadership style may exert a more direct and powerful influence of individual relational behavior and on group structure. Thank you for this interesting observation.

Comment 6: In the list of references, out of 73 literary sources, only 11 are dated for the last 5 years.

Response: I understand the observation and in fact, it is possible that I have left out of the article more current texts. However, this is explained by the fact that the relational mechanisms analyzed in this work correspond to classic themes within the field of social network analysis, which explains the -relative- antiquity of some of the references. However, I really consider that the literature reviewed is complete and sufficient to justify the theoretical relevance of the problem addressed.

Again, many thanks for the effort invested by the two reviewers to improve the quality of my work.

Best regards.
